# Enhancing Oil Recovery and Altering Wettability in Carbonate Reservoir Rocks through (3-Glycidoxypropyl) trimethoxysilane–SiO$_2$ Nanofluid Injection

Hochang Jang [1] and Jeonghwan Lee [2,*]

[1] Department of Energy Resources and Chemical Engineering, Kangwon National University, 346 Jungang-ro, Samcheok 25913, Republic of Korea; hcjang@kangwon.ac.kr

[2] Department of Energy and Resources Engineering, College of Engineering, Chonnam National University, 77 Yongbong-ro, Buk-gu, Gwangju 61186, Republic of Korea

* Correspondence: jhwan@jnu.ac.kr; Tel.: +82-33-530-1730

**Abstract:** This study analyzes the impact of injection condition design factors of (3-glycidoxypropyl) trimethoxysilane (GPTMS)–SiO$_2$ nanofluid on improving wettability and oil recovery through flotation and core flooding tests, respectively. Flotation tests were conducted to assess improvements in wettability that resulted from varying nanoparticle concentration, reaction time, and treatment temperature. The test results demonstrated that the hydrophilic sample ratio increased by up to 97.75% based on the nanoparticle reaction, confirming significant wettability improvement in all samples. Additionally, time-dependent fluid-flow experiments were conducted to validate oil recovery and rock–fluid interactions. In these experiments, for a 24-h reaction time, nanofluid injection caused a decrease in the maximum contact angle (43.4° from 166.5°) and a remarkable enhancement in the oil recovery rate by over 25%. Moreover, variations in contact angle and sample permeability were observed as the reaction time increased. Subsequently, the core flooding test revealed a critical reaction time of 24 h, maximizing oil recovery while minimizing permeability. Below this point in time, wettability improvement did not significantly enhance oil recovery. Conversely, beyond this threshold, additional adsorption due to particle aggregation decreased permeability, causing reduced oil recovery. Therefore, GPTMS–SiO$_2$ nanofluid can be utilized as an injection fluid to enhance oil recovery in high-temperature and high-salinity carbonate reservoirs.

**Keywords:** GPTMS; carbonate reservoir; enhanced oil recovery; flotation test

## 1. Introduction

Petroleum in carbonate reservoirs constitutes 60% of the global oil reserves and has significant potential for enhanced oil recovery (EOR) [1]. However, owing to the heterogeneity and oil-wet characteristics of carbonate formations, the effectiveness of conventional waterflooding, a secondary recovery method, is limited, resulting in lower oil recovery [2–4]. With the advancement of nanotechnology, its application in the oil and gas industry has expanded. In particular, the use of nanofluids in the field of EOR is increasing, offering the promise of improved oil recovery through various mechanisms, such as wettability alteration, structural disjoining pressure, and interfacial tension (IFT) reduction.

The primary mechanisms of nanoscale particles involve rock–fluid interactions and fluid–fluid interactions. In rock–fluid interactions, nanoscale particles are adsorbed onto the rock surface, displacing oil droplets and altering the wettability of the rock to render it more hydrophilic, thereby inducing oil recovery. However, pore deformation or plugging may occur during adsorption, leading to decreased permeability. As the degree of adsorption increases, oil recovery can be simultaneously enhanced by increasing hydrophilicity and reducing fluid mobility owing to decreased permeability. Therefore, a comprehensive anal-

ysis is essential to understand the effects of adsorption on both wettability and permeability and their implications for oil recovery.

Roustaei and Bagherzadeh [5] analyzed the EOR effect of silica nanoparticles (SNPs) on carbonate rock samples. In their observations, the oil recovery rate varied with the nanoparticle concentration. Al-Anssari et al. [6] investigated the impact of $SiO_2$-based nanofluids on the wettability of carbonate rocks. They determined the minimum reaction time required for a sample to transition from hydrophobic to hydrophilic. Chandio et al. [7] used hydrophilic silica nanofluids to determine the effects of nanofluid salinity and particle concentration on oil recovery and stability. They found that wettability alteration and reduced interfacial tension maximized oil recovery at a salinity level of 20,000 ppm and a nanoparticle concentration of 0.05 wt.%. Notably, typical salinity levels in carbonate reservoirs are approximately 90,000 ppm, indicating a high brine environment. This may affect the dispersion stability of nanoparticles, potentially leading to reduced efficiency in nanoparticle injection methods and hindering reservoir productivity. Iijima et al. [8] and Yang and Liu [9] polymerized silica nanoparticles with silane and confirmed their high dispersion stability. Ranka et al. [10] also demonstrated stable dispersion over a week in environments with temperatures up to 90 °C and salinity levels of 12 wt.% by binding cationic polymers to silica nanoparticles. Jang et al. [11] manufactured stable nanofluids using a silane coupling agent called GPTMS ((3-glycidoxypropyl)trimethoxysilane) and silica nanoparticles at 20 wt.% salinity and 90 °C. They also confirmed that the nanofluid had a positive effect on the wettability of carbonate rock samples.

The synthesis of nanoparticles using chemical methods and subsequent polymerization can enhance their dispersion stability, even in high-temperature and high-salinity environments [12–14]. These nanoparticles can be injected into carbonate reservoirs to aid oil recovery. When nanoparticles undergo functionalization, the characteristics of the resulting material and their interaction with the reservoir environment can be influenced, affecting both fluid–fluid and fluid–rock interactions. To optimize nanofluid flow using injection methods, it is crucial to characterize the produced nanofluid. Specifically, it is essential to analyze the impact of nanofluid reactivity (adsorption capacity) on wettability alteration, which is one of the main mechanisms. This can significantly affect the oil recovery rate by altering the wettability. Mazen et al. [15] employed the contact angle measurement method for wettability analysis and varied the conditions for various rock samples to derive the results. In addition, flotation tests involve creating rock particles in powder form and conducting experiments in which the wetting state of the rock surfaces affects the buoyancy and settling mechanisms of the experimental fluid. Sadeghi et al. [16] provided insights into the wettability of rock particles and the characteristics of rocks that influenced this test. However, considering temperature variations and reaction times in the context of nanofluid reactivity is important because they are essential factors that need to be adequately addressed in fluid flow experiments. These factors can significantly impact the effectiveness of the injection method and, consequently, oil recovery.

This study assessed the potential of GPTMS–$SiO_2$ nanofluids to enhance wettability in carbonate reservoirs. Flotation tests were conducted to analyze key factors—nanoparticle concentration, reaction temperature, and time—that influence wettability. The quantification of contact-angle variations highlighted the impact of the nanofluid. These findings underscore the potential of GPTMS–$SiO_2$ nanofluids to enhance wettability in carbonate reservoirs. Furthermore, fluid flow experiments, considering the nanofluid's reaction time, were conducted on both crude oil and core plug samples from an operating oil field. These experiments analyzed pressure differentials, recovery rates, and contact-angle changes induced by nanofluid injection, providing crucial insights into the applications of GPTMS–$SiO_2$ nanofluids for improving fluid flow in carbonate reservoirs.

## 2. Materials and Methods

The American Petroleum Institute (API) brine was used as the simulated formation water, consisting of NaCl (80 g/L) and $CaCl_2$ (20 g/L) at a salinity of 100,000 mg/L. NaCl

(99% purity) and $CaCl_2$ (93% purity) were purchased from Daejung Chemicals & Metals, Republic of Korea, and used without further purification. GPTMS and LUDOX® TMA colloidal SNPs were purchased from Sigma-Aldrich Korea (Seoul, Korea) and used without further purification. GPTMS was employed as a silane coupling agent, consisting of a trimethoxy group and epoxy, and primarily has an $R\text{-}Si(OR')_3$ structure for the surface modification of the nanoparticles. The SNPs were added to 34 wt.% silica suspension in $H_2O$ (pH 4–7), and each particle, with a nominal diameter of 20 nm, has an approximate surface area of 140 $m^2/g$. Kerosene (for flotation tests) and crude oil (for core flooding tests) were utilized, and the properties of these oils are listed in Table 1. The rock samples were composed of Indiana limestone ($CaCO_3$, 97.07%) and Silurian dolomite ($Ca(Mg)CO_3$, 93.81%) purchased from Kocurek Industries, Inc. (Caldwell, TX, USA). For experimental purposes, the rock samples were prepared in powder form, sized between 44 and 74 microns (200 and 325 mesh), as detailed in Table 2. Furthermore, the core flooding experiments used core plugs obtained from rock samples collected from an operational field. The conventional rock properties of the carbonates used for nanofluid injection are listed in Table 3.

**Table 1.** Crude oil properties.

| API Gravity, °API | Specific Gravity, Fraction | Viscosity at 60 °C, mPa·s | Acid Number, mg/g KOH | Base Number, mg/g KOH |
|---|---|---|---|---|
| 32.7 | 0.861 | 5.75 | 0.1 | 1.1 |

**Table 2.** Experimental conditions for flotation and core flooding tests.

| | Flotation Test | Core Flooding Test |
|---|---|---|
| Litho type | Carbonate (Indiana limestone and Silurian dolomite) | Low density, high phi carbonate (core plug drilled field) |
| Sample type | Powder (44–74 micron) | Core plug (1.5 inch diameter) |
| Test oil | Kerosene | Crude oil |

**Table 3.** Carbonate rock properties for nanofluid injection performance.

| No. | Litho-Type | Diameter, cm | Length, cm | Permeability, md | Porosity, % | Pore Volume, cc |
|---|---|---|---|---|---|---|
| #01 | | 3.75 | 6.74 | 38.1 | 28.1 | 20.9 |
| #02 | Low density, high phi carbonate | 3.78 | 6.51 | 9.0 | 27.3 | 19.9 |
| #03 | | 3.77 | 6.53 | 37.0 | 30.3 | 22.1 |
| #04 | | 3.75 | 6.71 | 10.0 | 26.7 | 19.8 |

### 2.1. $SiO_2$ Nanofluid Preparation

Sandstone rocks have a uniform pore system and are homogeneous, whereas carbonate rocks have a complex pore system consisting of macro- and micro-pores. When injecting nanoparticles of the same size into both reservoirs, the pores are easily plugged by nanoparticles, and this effect is further amplified by particle aggregation. Therefore, the particle-size sensitivity was compared between sandstone and carbonate reservoirs for the injection of nanofluids [17]. Additionally, preventing particle aggregation under reservoir conditions was important. Achieving colloidal stability under reservoir conditions is crucial for recovering residual oil from carbonate reservoirs, necessitating tiny nanoparticles. GPTMS–$SiO_2$ comprises surface-modified silica, which resists high-concentration electrolytes and elevated temperatures. Additionally, it has proven effective in altering the wettability of carbonates [11]. To prepare the GPTMS–$SiO_2$ nanofluid (as illustrated in Figure 1), we added 1.0 mmol/g of GPTMS to the SNPs in deionized water to achieve a pH range of 6–7, resulting in a 10 wt.% SNP concentration. Magnetic stirring facilitated the addition of GPTMS to the diluted SNPs, followed by stirring for 1 h and heating at 70 °C

for 24 h. This process involved GPTMS hydrolysis and grafting onto the SNP surfaces, enhancing the colloidal stability of the nanofluid. The hydrolysis reaction is as follows [18]:

$$RSI(OCH_3)_3 \overset{H_2O}{\Rightarrow} RSI(OH)_3 + 3CH_3OH. \tag{1}$$

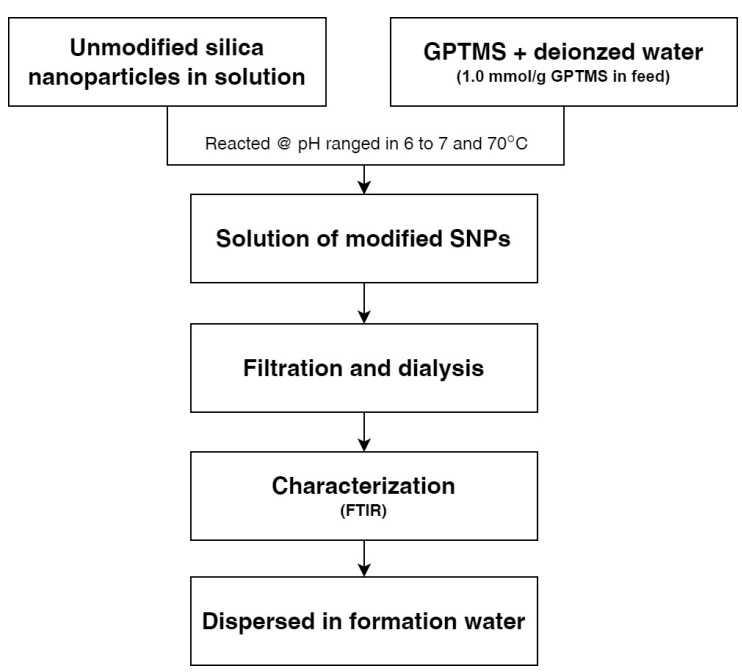

**Figure 1.** Preparation process for the nanofluid composed of surface-modified nanoparticles.

At 70 °C, OR′ covalently bonds to free hydroxyl groups on SNP surfaces, modifying surface characteristics. Rather than reducing the electrostatic repulsion, this process enhances the repulsive forces and kinetically stabilizes the GPTMS–SiO$_2$ nanofluid under high electrolyte and temperature conditions [11]. Subsequently, the solution of modified SNPs was filtered using cellulose esters and purified via dialysis against deionized water to remove ungrafted SNPs. Following this process, the impact of the GPTMS on grafting was assessed via FTIR at Kangwon National University's central laboratory in Samcheok, Republic of Korea. Then, the grafted product was dispersed in simulated formation water.

### 2.2. Flotation Experiment for Wettability Alteration Test

To conduct the flotation test (Figure 2), limestone and dolomite rocks were crushed into powder sized between 44 and 74 microns using crushers and ball mills. To simulate the initial conditions and create an oil-wet state, 2 g of the rock sample was introduced into a conical tube, and 10 g of oil was added. Subsequently, a two-day reaction at 75 °C occurred in an oven, and 20 g of the GPTMS–SiO$_2$ nanofluid was introduced. Furthermore, the experiments considered operational factors that could be affected, including a nanoparticle concentration of 0.1–3 wt.%, reaction temperatures ranging from 30 to 90 °C, and reaction times from 6 h to 72 h, as listed in Table 4. Once the reaction was complete, the samples underwent moisture removal by drying after removing the sample floating at the top. The wettability alteration was observed by comparing the percentage of water-wet rock particles at the end of the flotation test (%WW) of each experimental result and was quantified using Equation (2).

$$\%WW = (M_{ww} - M_r)/S_t \times 100, \tag{2}$$

where M$_{ww}$ is the mass of the clean and dry water-wet rock; M$_r$ is the mass of the remaining nanoparticles and salts after drying; and S$_t$ is the total solid weight in the initial feed.

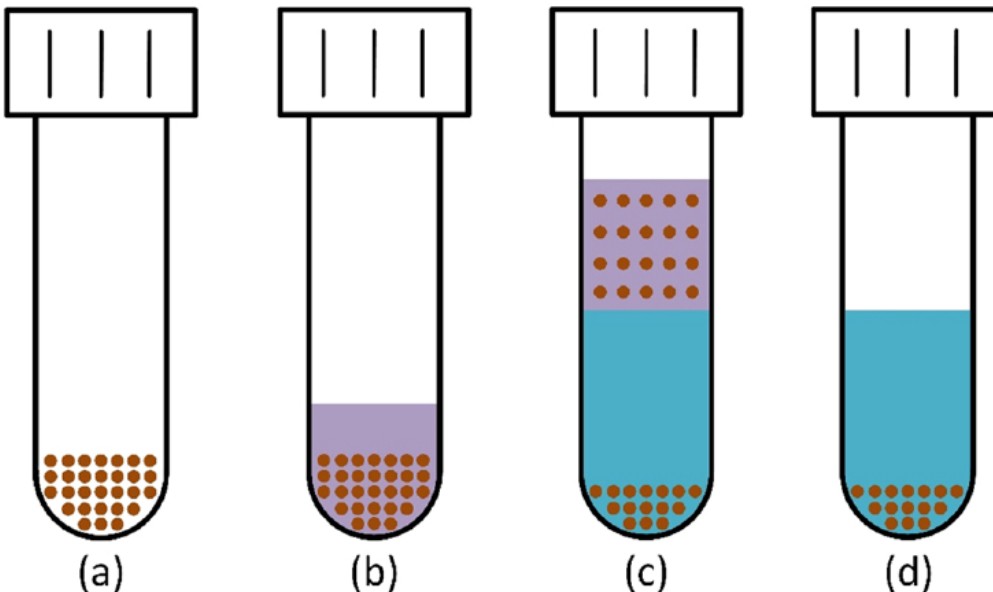

**Figure 2.** Flotation test processes: (**a**) Addition of the rock powder into a test tube. (**b**) Oil-aging of the rock powder in an oven for 2 days at 75 °C. (**c**) Vigorous mixing of the mixture with nanofluid and then placing in an oven for 1 day, causing oil-wet particles to float, whereas water-wet particles settle. (**d**) Removal of the oil phase and oil-wet particles, leaving the water-wet particles for cleaning, drying, and weighing (reproduced with permission from Sadeghi et al. [16]).

**Table 4.** Experimental conditions for wettability alteration by flotation testing.

| Parameters | Value |
|---|---|
| Amount of GPTMS in the feed | 1 mmol/g $SiO_2$ |
| Particle concentration (wt.%) | 0, 0.1, 0.5, 1, 2, 3 |
| Reaction time with nanoparticle (h) | 6, 12, 24, 48, 72 |
| Temperature (°C) | 30, 45, 60, 75, 90 |

*2.3. Core Flooding Experiment for Performance Evaluation of Nanofluid*

Figure 3 shows the core flooding system used for the aging process. Plug-core samples were saturated with API brine and crude oil, and the core samples were then displaced by oil until no more brine could be recovered. The plug core samples were placed in a core-aging cell, and the aging cells were pressurized using an injection pump. Subsequently, the aged cells were placed in an oven and maintained at 20,684 kPa and 90 °C for two weeks. Next, the core plugs were displaced using six pore volumes (PV) of oil. Finally, the core plugs were stored at room temperature for 10 days. Subsequently, the contact angle in the inlet of the core plug was measured using the sessile drop method with an Attension® Theta Lite by Biolin Scientific. Next, the nanofluid injection process was implemented as follows: First, 3 PV of brine are injected for waterflooding. Then, 1 PV of the nanofluid was injected into the core, and the treatment time was 12–72 h. Post-waterflooding was then conducted with 3 PV of brine. In all cases, the flow rate is set to 0.5 mL/min, and the system temperature remains at 80 °C. During the experiment, the inlet and outlet pressures were obtained using pressure sensors, and the production fluid was sampled every 3 mL. The experimental design parameters of the core flooding test are presented in Table 5. After the core flooding experiment, the contact angle in the inlet part of the core plug was measured again to compare the pre- and post-treatments with nanofluids during the core flooding test.

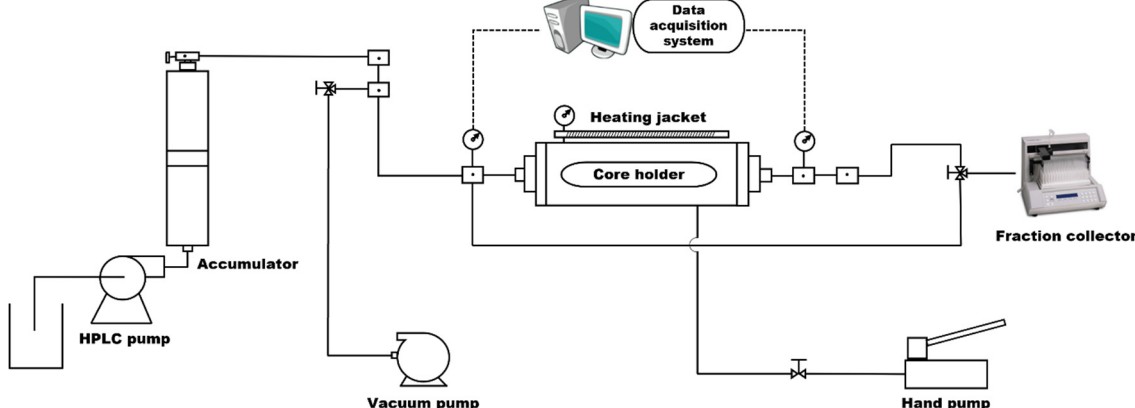

**Figure 3.** Schematic of the core flooding system for the EOR process using nanofluid.

**Table 5.** Experimental design parameters for the core flooding test.

| Parameters | Value |
| --- | --- |
| Amount of GPTMS in the feed | 1 mmol/g $SiO_2$ |
| Particle concentration (wt.%) | 1.0 |
| Salinity (salt concentration) | 100,000 mg/L (NaCl:$CaCl_2$, 8:2) |
| Reaction time with nanoparticle (h) | 12, 24, 48, 72 |
| Temperature (°C) | 80 |
| Flow rate (mL/min) | 0.5 |

## 3. Results and Discussion

### 3.1. Nanoparticle Dispersion Characteristics

As shown in Figure 4, a peak was observed at 2950 cm$^{-1}$ in the spectra after surface modification, corresponding to the symmetric C–H stretching vibrations attributed to the hydrocarbon chain in GPTMS [19]. The C–H stretching vibration indicates silane grafting on the silica surface. When GPTMS was added to the feed at concentrations of 1.0 mmol/g, symmetric C–H stretching vibrations were observed in the FTIR spectrum, indicating the completion of the grafting [20].

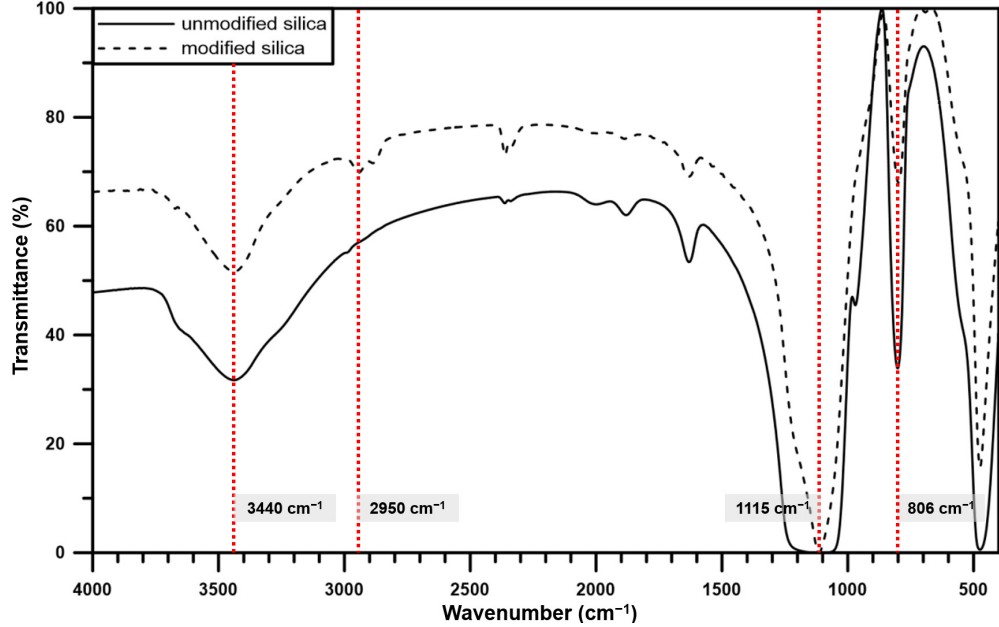

**Figure 4.** FTIR spectra for unmodified and surface-modified SNPs.

### 3.2. Floatation Test Results

Figure 5 shows the results of the flotation tests. After aging, oil-wet samples were produced; depending on the role of the nanofluid, they were separated into oil- and water-wet powders. The light oil-wet powder floated, whereas the heavy water-wet powder settled at the bottom. The results for the samples under varying nanoparticle concentrations, reaction times, and reaction temperatures were difficult to confirm visually; however, they were separated into different phases.

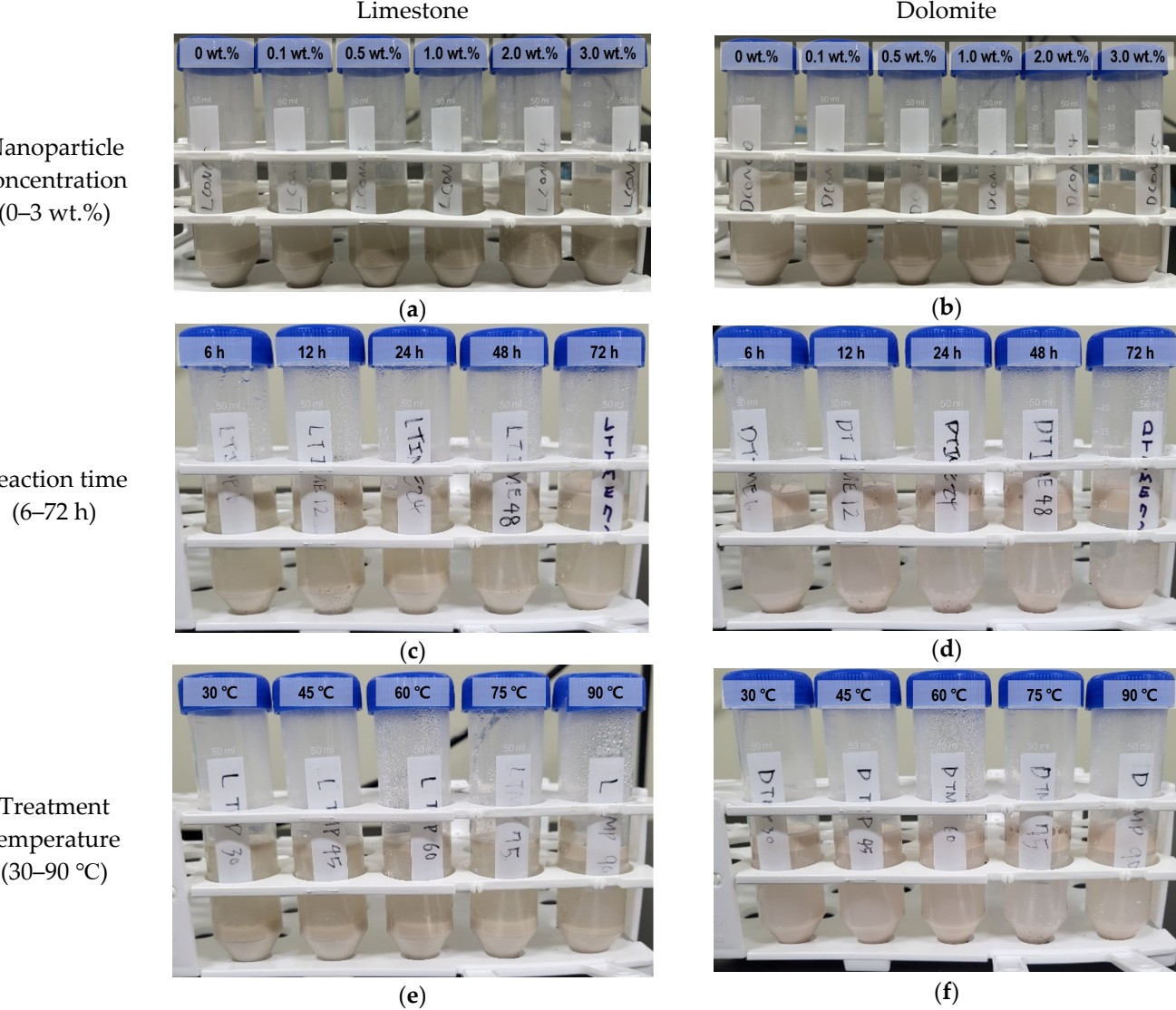

**Figure 5.** Floatation test results of rock powder with limestone (left) and dolomite (right). (**a**,**b**) Effects of nanoparticle concentrations. The oil phase and oil-wet particles are removed, leaving the water-wet particles. (**c**,**d**) Effects of reaction time. The mixture is vigorously mixed with brine and then placed in the oven for 1 day, causing the oil-wet particles to float, whereas the water-wet particles settle. (**e**,**f**) The effects of treatment temperature are similar to those of reaction time.

Particle concentration was set to 0–3 wt.% to examine its influence, whereas other variables, such as reaction time and temperature, were fixed at 24 h and 75 °C, respectively. Considering that the change in wettability is significantly affected by the reactivity of nanoparticles on the rock particle surface, an increase in particle concentration can potentially lead to an increase in reactivity. In addition, an increase in the fluid particle concentration can affect the stability of the nanoparticle dispersion. Figure 6 shows the values of %WW as a function of particle concentration. For limestone, the hydrophilic sample

exhibited a ratio of 52.3% in the absence of nanoparticles. With the reaction of nanoparticles, the %WW increased to 68–97.75%, indicating a significant change in wettability owing to the nanoparticle reaction. For dolomite, the base fluid exhibited approximately 73% of %WW. Furthermore, the ratio of the hydrophilic sample gradually increased from 83% to 98% with increasing nanoparticle concentration. Similarly, changes in the wettability of the oil-wet samples were observed. Limestone had a higher initial oil-wet ratio, whereas dolomite was more sensitive to the concentration of rock particles. Thus, an improvement in wettability occurs in both limestone and dolomite when they react with nanoparticle-containing fluids. Furthermore, the extent of improvement increases with dolomite concentration. Moreover, limestone showed wettability improvement regardless of concentration. This indicates that particle surface characteristics can influence reactivity, highlighting the need for particle concentration control based on the rock composition in the reservoir.

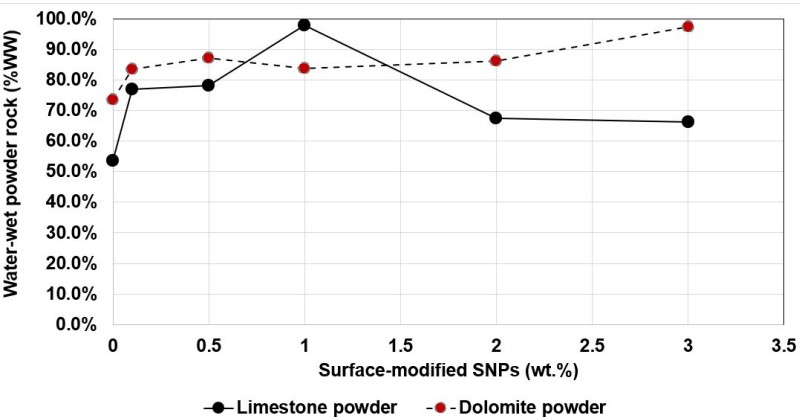

**Figure 6.** Effect of particle concentrations of SNPs on flotation test results.

To assess the effect of reaction time, the nanoparticle concentration was set to 1 wt.%, and the reaction temperature was fixed to 75 °C, with analyses conducted over 6–72 h. Figure 7 shows the values of %WW as a function of reaction time with nanofluid. In the case of limestone, slight variations in %WW were observed over time. However, all samples exhibited differences of less than 3% for dolomite. The impact of reaction time was not clear, and wettability improved irrespective of time. However, considering the distinctive characteristics of powdered samples with significantly higher surface areas than the actual core samples, the reactivity between nanoparticles and rock surfaces may be overstated. Hence, it is important to recognize the need for analyzing the results of core flooding experiments.

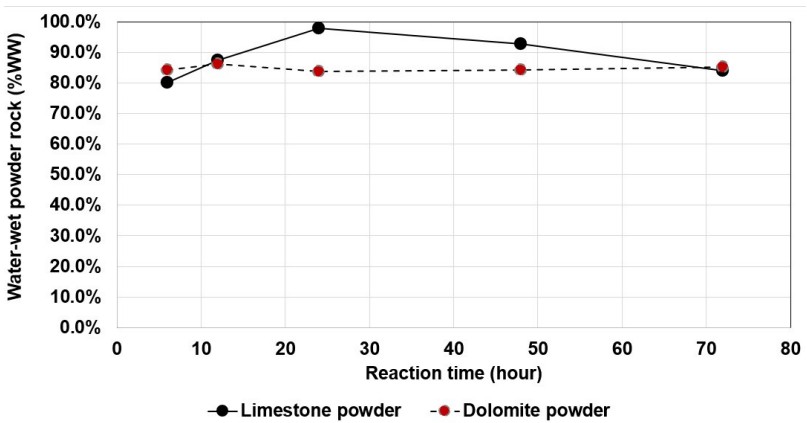

**Figure 7.** Effect of reaction time on flotation test results.

To examine the influence of temperature, the concentration of the nanofluid was fixed to 1 wt.%, and the reaction time was set to 24 h. Figure 8 shows the %WW as a

function of temperature. Both limestone and dolomite showed significant improvements in wettability under relatively low-temperature conditions, whereas in high-temperature environments, the %WW relatively decreased, indicating a reduction in wettability change. This phenomenon is associated with an increased tendency of nanoparticles to maintain a dispersed state in the base fluid rather than adsorb onto the rock surface as the temperature increases. In both cases, it was observed that, compared with the maximum %WW, the reaction samples at 90 °C exhibited differences of more than 10%.

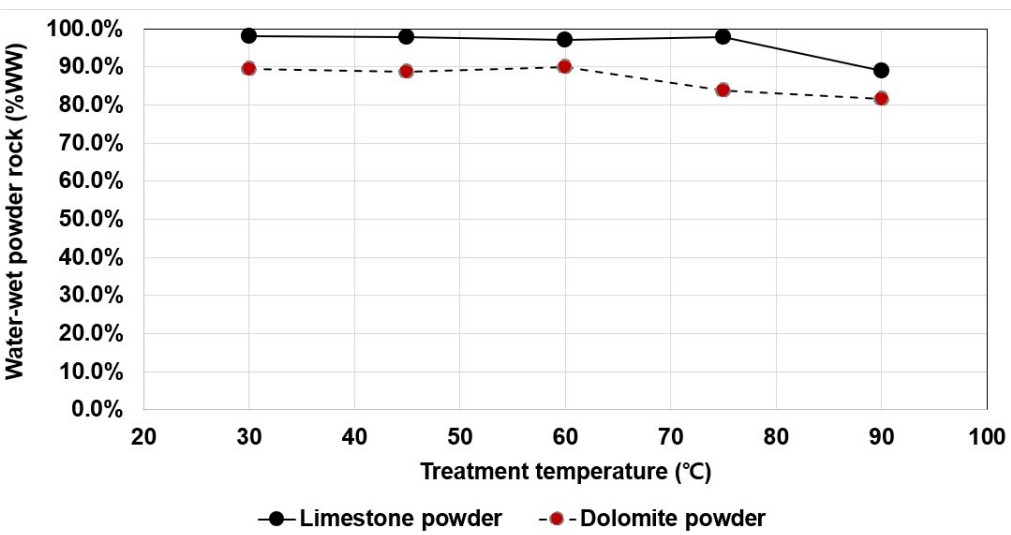

**Figure 8.** Effect of treatment temperature on flotation test results.

### 3.3. Results of the Core Flooding Test

The high reactivity of the powdered samples used in the flotation test suggests that they yielded results more rapidly than the interaction with actual rock samples during the fluid flow processes. Therefore, to confirm the influence of the reaction time more accurately, core flooding experiments were conducted with varying reaction times, along with examining the wettability improvement of nanoparticles, oil recovery, and changes in the properties of rock particles.

Waterflooding, nanofluid injection, and post-waterflooding of 3, 1, and 3 of PV, respectively, were performed after a controlled reaction time. Figure 9 shows the recovery and pressure drop across the core before and after PVI when the reaction time was 12 h. The oil recovery increased by approximately 45% during the waterflooding period, with an additional 10% increase in the remaining intervals. Among the series of injections, the pressure changes were found to stabilize within 20 psi, and the pressure drop in the later interval gradually increased. Moreover, it can be observed that the pressure difference between the early and late stages is not significant. The change in permeability before and after nanofluid injection was minimal; however, the increase in the recovery rate was also modest, indicating an inefficiency. Furthermore, the contact angle decreased slightly from 134.8° to 125.1° before and after the nanofluid reaction. The extent of this change was not substantial and was still ineffective in altering the wettability in an oil-wet state.

When the reaction time was 24 h, a pronounced increase in the recovery rate was observed (Figure 10). The stable pressure levels before and after waterflooding in the field increased from approximately 30 psi to 50 psi; however, the pressure difference was not significant. During the waterflooding period, 54.1% of the oil was recovered, followed by an additional 7.1% during the subsequent nanofluid injection. After the nanofluid reaction, 18.7% was recovered during the post-waterflooding period. It was confirmed that 25.8% in total was recovered by applying EOR methods after the waterflooding phase. The contact angle significantly decreased from 166.5° to 43.4°, demonstrating a substantial shift from strong oil-wet conditions to strong water-wet conditions on the surface.

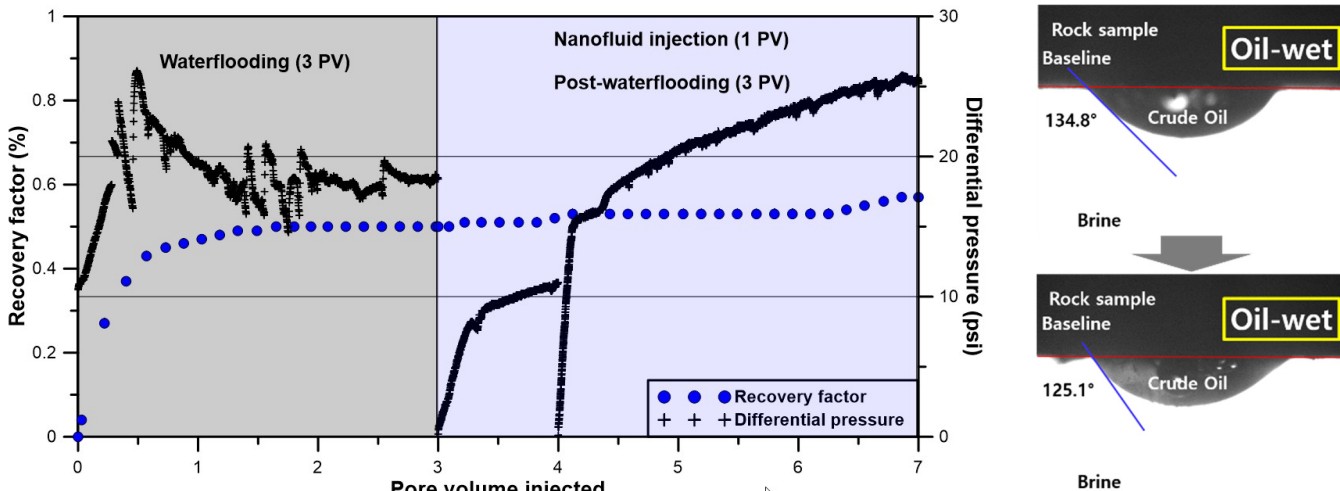

**Figure 9.** Oil recovery and differential pressure (**left**) during the core flooding test, and contact angles (**right**) in pre- and post-treatment at a reaction time of 12 h.

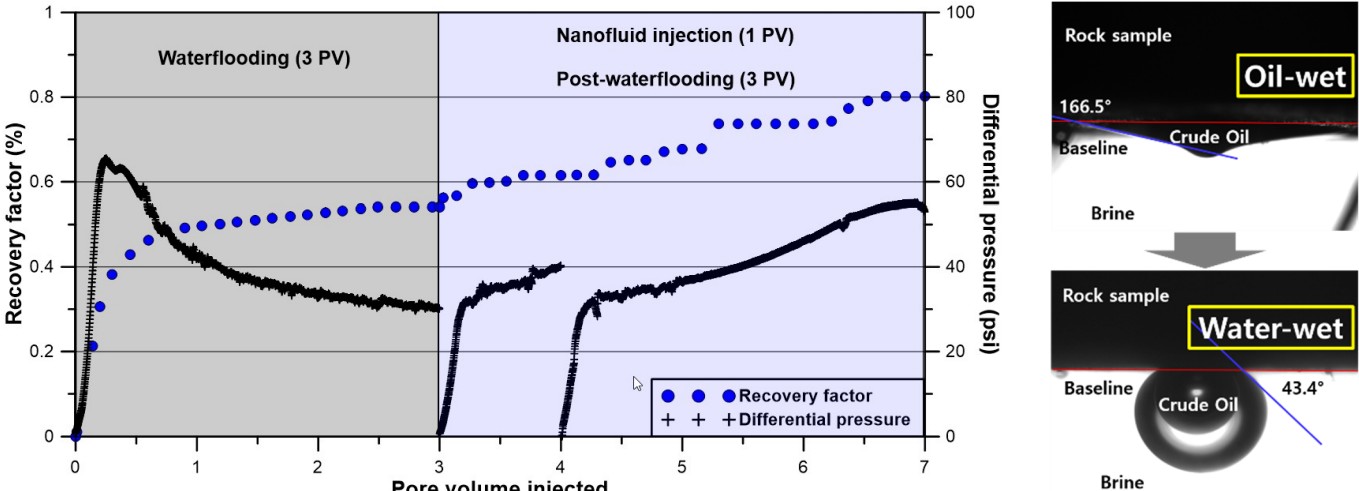

**Figure 10.** Oil recovery and differential pressure (**left**) during the core flooding test, and contact angles (**right**) in pre- and post-treatment at a reaction time of 24 h.

For a reaction time of 48 h, an additional oil recovery of 12.6% was confirmed after the waterflooding period, and the stable pressure levels before and after waterflooding in the field increased by approximately twofold, from 10 psi to 20 psi (Figure 11). The contact angle significantly decreased from 165.4° to 62.4°, indicating a substantial shift from strong oil-wet conditions to strong water-wet conditions on the surface.

At a reaction time of 72 h, the additional recovery rate was less than 10%, indicating a reduced increase in the recovery rate, as shown in Figure 12. In the water-flooding section, the pressure was approximately 23 psi; however, in post-waterflooding, the pressure did not stabilize and continued to increase throughout the injection of 3 PV. This indicates that nanoparticle aggregation was generated in the core plug by long-term exposure to nanofluids at high temperatures. In addition, it suggests that the reaction time is a sensitive parameter for altering rock properties such as wettability and permeability. The contact angle significantly decreased from 166.5° to 43.4°, confirming a substantial shift from strong oil-wet conditions to strong water-wet conditions on the surface.

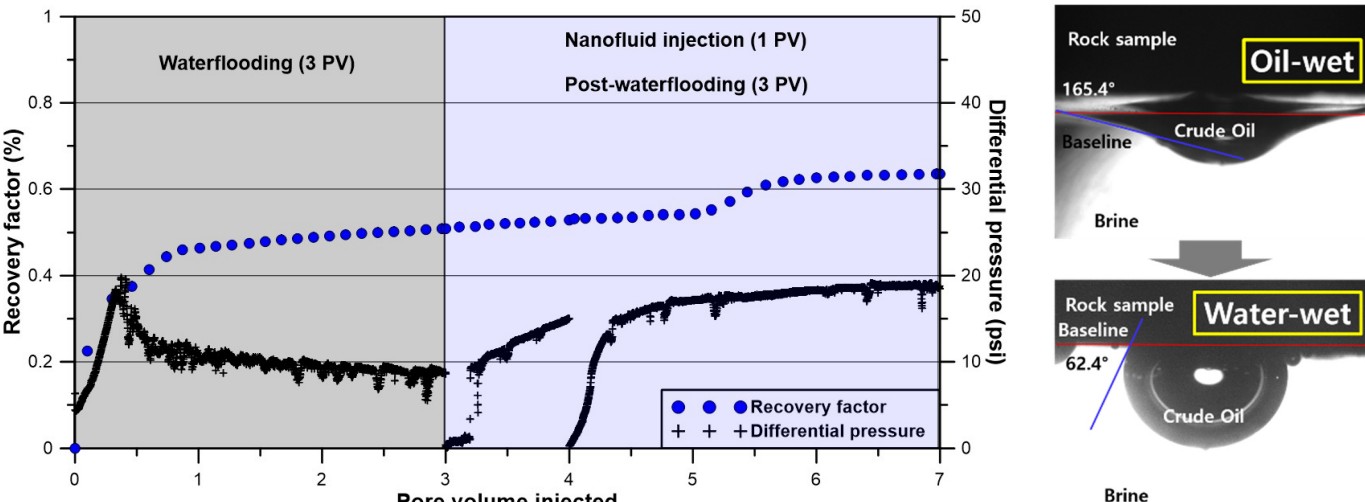

**Figure 11.** Oil recovery and differential pressure (**left**) during the core flooding test, and contact angles (**right**) in pre- and post-treatment at a reaction time of 48 h.

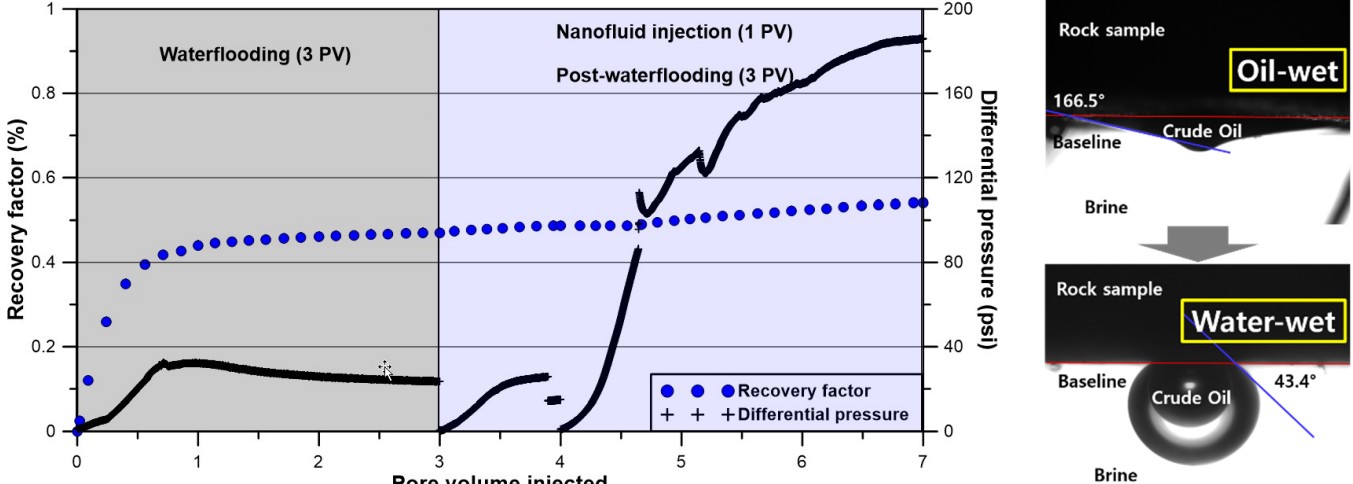

**Figure 12.** Oil recovery and differential pressure (**left**) during the core flooding test, and contact angles (**right**) in pre- and post-treatment after a reaction time of 72 h.

When the reaction time was short, the nanofluid did not sufficiently react on the rock surface, leading to reduced adsorption and permeability by the nanoparticles. However, owing to its low reaction rate, it may not significantly alter the wettability. Therefore, the role of fluid-to-fluid interactions and fluid-to-rock interactions (interpreted through changes in pressure and contact angle) is diminished. Fluid-to-fluid and fluid-to-rock reactions were observed when an appropriate reaction time was provided. Some permeability reduction and wettability changes occurred. These results also confirmed improved recovery rates, with a reaction time of 24 h being the best-case scenario in this experiment. As the reaction time increased, the permeability reduction effect, which is one aspect of the fluid-to-rock interaction, became more pronounced, leading to a decrease in oil recovery efficiency. Therefore, designing an appropriate reaction time is crucial for maximizing the effects of rock–fluid interactions in EOR. To achieve this, analyzing the reaction rates and their impact via a fluid-to-rock interaction analysis is necessary. Table 6 summarizes the experimental results of the core flooding test. The incremental oil recovery ranged from 7.2% to 26.1%, with varying reaction times. Considering the high residual oil in carbonate reservoirs after waterflooding, nanofluid injection with GPTMS–SiO$_2$ appears to have been successful for EOR. The contact-angle differences ranged from 9.7° to 123.1° with nanofluid treatment,

and wettability improvement was observed in all samples with reaction times exceeding 12 h. This implies that sufficient reaction time is necessary to enhance the wettability of a carbonate rock. Permeability exhibited a decrease with an increase in reaction time, with a permeability ratio of 0.78 at a 12 h reaction time and 0.11 at a 72 h reaction time, respectively.

**Table 6.** Experimental results with different reaction times.

| Sample No. | #01 | #02 | #03 | #04 |
|---|---|---|---|---|
| Incremental oil recovery | 7.2% | 26.1% | 12.6% | 7.1% |
| Permeability ratio | 0.78 | 0.64 | 0.48 | 0.11 |
| Contact angle differences | 9.7° | 123.1° | 103.0° | 107.1° |

### 4. Discussion

The GPTMS–SiO$_2$ nanofluid has been observed to exhibit changes in wettability through contact-angle variations on limestone and dolomite under high-salinity and high-temperature conditions, as confirmed in a previous study [11]. Furthermore, viscosity, a key factor in fluid flow processes, has been analyzed under enhanced oil recovery operating conditions [21]. In this study, we focused on enhancing the oil recovery potential of this nanofluid through its reactivity with carbonate rock formations. To achieve this, flotation tests were conducted to increase the specific surface area of rock samples and maximize the fluid–rock interaction [16]. Additionally, core flooding experiments were performed to represent fluid–rock reactions in the flowing path within actual carbonate reservoirs. Throughout the experiments, we observed that this nanofluid could increase the hydrophilicity of the rock samples, altering the rock surface from oil-wet to water-wet conditions under the overall experimental conditions. Notably, the composition of the rock had an impact on the performance of the nanofluid, including nanofluid concentration and reaction time, which are operational factors affecting fluid displacement in rocks. However, note that flotation tests, as previously explained, may not quantitatively represent the reactivity of rock–fluid interactions observed in actual fluid performance. Instead, they can be qualitatively described in terms of their effectiveness in improving wettability. Furthermore, the analysis of core flow experiments indicated that the contact angle no longer changes with increasing reaction, implying the existence of a critical range. However, the permeability reduction effect continued to increase. This implies that nanoparticle adsorption on the rock surface forms not a monolayer but a bilayer or more. Considering that changes in wettability and permeability are crucial factors determining the success of EOR, selecting a critical reaction time point is deemed highly important. Beyond a critical reaction time, in our case, that is, for 24 h, no further changes in the contact angle were observed, and the permeability decreased. Therefore, determining the optimal reaction time, known as the critical time, is necessary from the perspective of oil recovery efficiency. This suggests the need for research on rock–fluid interactions to effectively operate EOR methods using nanofluids.

### 5. Conclusions

This study analyzed the impact of surface-modified nanoparticles on wettability improvement and oil recovery rates in carbonate reservoirs. Flotation tests were conducted based on the nanoparticle concentration, reaction time, and treatment temperature. In addition, core flow experiments were conducted on field samples obtained from the oil field to investigate the impact of reaction time. The following results were obtained:

1.　The flotation test results showed that the hydrophilic sample ratio increased to 97.75%, confirming successful wettability improvement in the limestone and dolomite reservoirs. In the case of limestone, the results were consistent regardless of the particle concentration, whereas for dolomite, the degree of improvement increased with higher concentrations. Therefore, the composition of the reservoir rock must be considered when determining the concentration for nanofluid applications.

2. The core flow experimental results indicated that the additional oil recovery through nanofluid injection and subsequent waterflooding reached a maximum of 26.1%. Adjusting the reaction time resulted in an incremental oil recovery ranging from 7.2% to 26.1%. The contact-angle differences ranged from 9.7° to 123.1° with nanofluid treatment, and wettability improvement was observed in all samples with reaction times exceeding 12 h. Permeability exhibited a decrease with an increase in reaction time. Specifically, the permeability ratio was 0.78 for a 12-h reaction time and 0.11 for a 72-h reaction time.

3. With increasing reaction time, wettability improved, and nanoparticle aggregation also occurred, leading to reduced permeability and decreased oil recovery. In our case, the critical reaction time was 24 h to maximize oil recovery while minimizing permeability reduction. Below this limit, the wettability improvement did not facilitate oil recovery enhancement. In contrast, additional adsorption owing to particle aggregation decreased permeability beyond this limit, thereby reducing oil recovery.

4. To maximize the efficiency of the EOR method using GPTMS–$SiO_2$ nanofluid, it is crucial to control the reactivity between the rocks and nanofluids by adjusting the rock composition, nanoparticle concentration, and reaction time.

**Author Contributions:** H.J.: conceptualization, methodology, writing—original draft preparation, visualization, and validation. J.L.: supervision, writing—review and editing, and validation. All authors have read and agreed to the published version of the manuscript.

**Funding:** This work was supported by (1) the National Research Foundation of Korea (NRF) grant funded by the Korean government (MSIT) (No. 2020R1F1A1048182), (2) the Energy and Mineral Resources Development Association of Korea (EMRD) grant funded by the Korean government (MOTIE) (2021060001, Data science-based oil/gas exploration consortium) and (3) the "Regional Innovation Strategy (RIS)" through the National Research Foundation of Korea (NRF) funded by the Ministry of Education (MOE) (2022RIS-005).

**Institutional Review Board Statement:** Not applicable.

**Informed Consent Statement:** Not applicable.

**Data Availability Statement:** Data will be made available on request.

**Conflicts of Interest:** The authors declare that they have no known competing financial interests or personal relationships that could have appeared to influence the work reported in this paper.

## Nomenclature

List of Abbreviations

| | |
|---|---|
| API | American petroleum institute |
| SNP | Silica nanoparticle |
| GPTMS | (3-glycidoxypropyl)trimethoxysilane |
| EOR | Enhanced oil recovery |
| FTIR | Fourier transform infrared spectroscopy |
| $R\text{-}Si(OR')_3$ | One organic group, in addition to three alkoxy groups |
| $H_2O$ | Water |
| $CH_3OH$ | Methanol |
| $CaCO_3$ | Calcium carbonate (limestone) |
| NaCl | Sodium chloride |
| $Ca(Mg)CO_3$ | Calcium magnesium carbonate (dolomite) |
| $CaCl_2$ | Calcium chloride |

List of symbols

| | |
|---|---|
| %WW | percentage of water-wet rock particles |
| $M_{ww}$ | Mass of the cleaned and dried water-wet rock |
| $M_r$ | Mass of the remaining nanoparticles and salt after drying |
| $S_t$ | Total solid weight in the initial feed |

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
