# Peer review of "Enhancing Oil Recovery and Altering Wettability in Carbonate Reservoir Rocks through (3-Glycidoxypropyl)trimethoxysilane–SiO2 Nanofluid Injection"

_applsci, doi:10.3390/app131911105_

Round 1

Reviewer 1 Report

1- It needs to give some physical significance in abstract section to gain the interest of readers.

2- The novelty of the paper should be clearly studied in the last paragraph of the introduction section.

3- The conclusion section should be enriched by adding some quantitative outcomes.

4- what is the source of table 1 ? 

5- please , provide more explanation to figure 1 

6- To inject nanofluids into carbonate reservoirs, the particle size sensitivity was compared to sandstone reservoirs"  can you explain in more details 

7- More physical explanations should be added to the results and discussion section.

8- Some grammatical errors and typos can be seen. Author should proofread the whole manuscript and correct it.

9- cite more recent references

Minor editing of English language required

Author Response

Amendments in response to the comments from Reviewer #1:

Thank you for appreciating our work. We thank reviewers for the thoughtful comments, which improved our manuscript. We hope our response has properly addressed the points raised in this review.

  1. It needs to give some physical significance in abstract section to gain the interest of readers.

Response: In response to the reviewer’s comment, we thoroughly revised and added the sentences in the abstract section as follows:

  • In these experiments, for a 24 h reaction time, nanofluid injection caused a decrease in the maximum contact angle (43.4° from 166.5°) and remarkable enhancement in the oil recovery rate by over 25%. Moreover, variations in contact angle and sample permeability were observed as the reaction time increased. Subsequently, the core flooding test revealed a critical reaction time of 24 h, maximizing oil recovery while minimizing permeability. Below this time, wettability improvement did not significantly enhance oil recovery. Conversely, beyond this threshold, additional adsorption due to particle aggregation decreased permeability, causing reduced oil recovery. Therefore, GPTMS-SiO2 nanofluid can be utilized as injection fluid to enhance oil recovery in high-temperature and high-salinity carbonate reservoirs.
  •  
  1. The novelty of the paper should be clearly studied in the last paragraph of the introduction section.

Response: In response to the reviewer’s comment, we thoroughly revised and condensed the last paragraph of the introduction section, focusing on the novelty of the paper.

  • Line 88: This study assessed the potential of GPTMS-SiO2 nanofluids to enhance wettability in carbonate reservoirs. Flotation tests were conducted to analyze key factors—nanoparticle concentration, reaction temperature, and time—that influence wettability. The quantification of contact-angle variations highlighted the impact of the nanofluid. These findings underscore the potential of GPTMS-SiO2 nanofluids in enhancing wettability in carbonate reservoirs. Furthermore, fluid flow experiments, considering the nanofluid's reaction time, were conducted on both crude oil and core plug samples from an operating oil field. These experiments analyzed pressure differentials, recovery rates, and contact-angle changes induced by nanofluid injection, providing crucial insights into the applications of GPTMS-SiO2 nanofluids for improving fluid flow in carbonate reservoirs.

  1. The conclusion section should be enriched by adding some quantitative outcomes.

Response: In response to the reviewer’s comment, we have added some quantitative outcomes as follows:

  • Line 396: The core flow experimental results indicated that the additional oil recovery through nanofluid injection and subsequent waterflooding reached a maximum of 26.1%. Adjusting the reaction time resulted in an incremental oil recovery ranging from 7.2% to 26.1%. The contact-angle differences ranged from 9.7° to 123.1° with nanofluid treatment, and wettability improvement was observed in all samples with reaction times exceeding 12 h. Permeability exhibited a decrease with an increase in reaction time. Specifically, the permeability ratio was 0.78 for a 12 h reaction time and 0.11 for a 72 h reaction time.

  1. what is the source of table 1 ?

Response: Owing to confidentiality concerns, the authors have not disclosed the source of the crude oil, which is why the details about its origin were not provided in the manuscript.

  1. please , provide more explanation to figure 1

Response: In response to the reviewer’s comment, we have provided additional explanation of Figure 1 as follows:

  • Line 134: (pH control) To prepare the GPTMS-SiO2 nanofluid (as illustrated in Figure 1), we added 1.0 mmol/g of GPTMS to the SNPs in deionized water to achieve a pH range of 6–7, resulting in a 10 wt.% SNP concentration.
  • Line 143: (Filtration and dialysis) Subsequently, the solution of modified SNPs was filtered using cellulose esters and purified through dialysis against deionized water to remove ungrafted SNPs.

  1. To inject nanofluids into carbonate reservoirs, the particle size sensitivity was compared to sandstone reservoirs" can you explain in more details

Response: In response to the reviewer’s comment, we have added explanation as follows:

  • Line 123: Sandstone rocks have a uniform pore system and are homogeneous, whereas carbonate rocks have a complex pore system consisting of macro- and micro-pores. When injecting nanoparticles of the same size into both reservoirs, the pores are easily plugged by nanoparticles, and this effect is further amplified by particle aggregation. Therefore, the particle-size sensitivity was compared between sandstone and carbonate reservoirs for the injection of nanofluids [17]. Additionally, preventing particle aggregation under reservoir conditions was important.
  1. More physical explanations should be added to the results and discussion section.

Response: In response to the reviewer’s comment, we have added some quantitative outcomes as follows:

  • Line 343: Table 6 summarizes the experimental results of the core flooding test. The incremental oil recovery ranged from 7.2% to 26.1%, with varying reaction times. Considering the high residual oil in carbonate reservoirs after waterflooding, nanofluid injection with GPTMS-SiO2 appears to have been successful for EOR. The contact-angle differences ranged from 9.7° to 123.1° with nanofluid treatment, and wettability improvement was observed in all samples with reaction times exceeding 12 h. This implies that sufficient reaction time is necessary to enhance the wettability of a carbonate rock. Permeability exhibited a decrease with an increase in reaction time, with a permeability ratio of 0.78 at a 12 h reaction time and 0.11 at a 72 h reaction time, respectively.
  • Line 372: Furthermore, the analysis of core flow experiments indicated that the contact angle no longer changes with increasing reaction, implying the existence of a critical range. However, the permeability reduction effect continued to increase. This implies that nanoparticle adsorption on the rock surface forms not a monolayer but a bilayer or more. Considering that changes in wettability and permeability are crucial factors determining the success of EOR, selecting a critical reaction time point is deemed highly important.

  1. Some grammatical errors and typos can be seen. Author should proofread the whole manuscript and correct it.

Response: Thank you for bringing this to our attention. Following your suggestions, we have thoroughly checked the entire manuscript for grammatical and language-related errors and have made corrections accordingly. Moreover, the manuscript was proofread by a professional English language editor to ensure that there were no language-related errors. Additionally, we have attached the editing certificate provided by Editage.

  1. cite more recent references

Response: In response to the reviewer’s comment, we have added the citations as follows:

  1. Arain, Z.-U.-A.; Al-Anssari, S.; Ali, M.; Memon, S.; Bhatti, M.A.; Lagat, C.; Sarmadivaleh, M. Reversible and irreversible adsorption of bare and hybrid silica nanoparticles onto carbonate surface at reservoir condition. Petroleum 2020, 6(3), 277–285, doi:10.1016/j.petlm.2019.09.001.
  2. Mogensen, K.; Masalmeh, S. A review of EOR techniques for carbonate reservoirs in challenging geological settings. J. Pet. Sci. Eng. 2020, 195, 107889, doi:10.1016/j.petrol.2020.107889.
  3. Ali, J.A.; Kolo, K.; Manshad, A.K.; Stephen, K.D. Emerging applications of TiO2/SiO2/poly(acrylamide) nanocomposites within the engineered water EOR in carbonate reservoirs. J. Mol. Liq. 2021, 322, 114943, doi:10.1016/j.molliq.2020.114943.
  4. Gbadamosi, A.O.; Junin, R.; Manan, M.A.; Yekeen, N.; Agi, A.; Oseh, J.O. Recent advances and prospects in polymeric nanofluids application for enhanced oil recovery. J. Ind. Eng. Chem. 2018, 66, 1–19, doi:10.1016/j.jiec.2018.05.020.
  5. Machale, J.; Majumder, S.K.; Ghosh, P.; Sen, T.K.; Saeedi, A. Impact of mineralogy, salinity, and temperature on the adsorption characteristics of a novel natural surfactant for enhanced oil recovery. Chem. Eng. Commun. 2022, 209, 143–157, doi:10.1080/00986445.2020.1848820.
  6. Song, C.; Jang, H.; Lee, J. Synthesis and dispersion stability of seawater-based nano-smart water for application in high-temperature and high-salinity conditions. Colloids Surf. A: Physicochem. Eng. Asp. 2023, 674, 131910, doi:10.1016/j.colsurfa.2023.131910.

Reviewer 2 Report

In this paper, the effects of surface modified nanoparticles on the wettability and oil recovery of carbonate reservoir are analyzed through flotation and core flooding experiments. It is pointed out that the design of appropriate reaction time is crucial to improve the effect of rock-fluid interaction in EOR. The article is rich in content and rigorous in thinking. However, I still hope that the author can make the following modifications:

What does the author mean by "Viscosity at 60, mPa∙s" in Table 1?

The names in Table 2 and Table 3 are the same, so it is suggested that the author check the relevant contents.

In Figure 1, the flow arrow overlaps with the content, suggesting the author to optimize the relevant content.

It is suggested to optimize the layout of tables and pictures.

The relevant contents of (a) and (b) in Figure 5 are blocked, and it is recommended to change the relevant pictures. If an unobstructed picture is not available, it is recommended to indicate the nanoparticle concentration used for each sample.

Please check whether the sentences and words in the text are correct, such as whether ' Coreflooding ' should be changed to ' Core flooding '.

In order to highlight the main strengths of your research, extensive literature research should be conducted to shed light on important issues in oil extraction, some work in this area includes: The non-plane initiation and propagation mechanism of multiple hydraulic fractures in tight reservoirs considering stress shadow effects. Engineering Fracture Mechanics; Exploring the influence of rock inherent heterogeneity and grain size on hydraulic fracturing using discrete element modeling. International Journal of Solids and Structures.

Author Response

Amendments in response to the comments from Reviewer #2:

In this paper, the effects of surface modified nanoparticles on the wettability and oil recovery of carbonate reservoir are analyzed through flotation and core flooding experiments. It is pointed out that the design of appropriate reaction time is crucial to improve the effect of rock-fluid interaction in EOR. The article is rich in content and rigorous in thinking. However, I still hope that the author can make the following modifications:

Response: We are grateful to you for appreciating our work and providing valuable feedback. We have revised the manuscript as per your suggestions. The pointwise response to your comments is provided below.

  1. What does the author mean by "Viscosity at 60, mPa∙s" in Table 1?

Response: Following the reviewer’s comment, we modified the term to 'Viscosity at 60 ℃, mPa∙s'.

  1. The names in Table 2 and Table 3 are the same, so it is suggested that the author check the relevant contents.

Response: In response to the reviewer’s comment, we accurately changed the caption of Table 2 to ' Table 2. Experimental conditions for flotation and core flooding tests.'

  1. In Figure 1, the flow arrow overlaps with the content, suggesting the author to optimize the relevant content.

Response: In accordance with the reviewer’s comments, we have accurately revised the procedure and corrected the flow arrows in Figure 1.

  1. It is suggested to optimize the layout of tables and pictures.

Response: In response to the reviewer’s comments, we have thoroughly revised the tables and pictures throughout the manuscript as follows:

  • Table 1: corrected units
  • Table 2: revised caption
  • Figure 1: updated procedure
  • Figure 5: added experiment information
  • Figures 9-11: revised left-side pictures

  1. The relevant contents of (a) and (b) in Figure 5 are blocked, and it is recommended to change the relevant pictures. If an unobstructed picture is not available, it is recommended to indicate the nanoparticle concentration used for each sample.

Response: Unfortunately, the samples were disposed of after the flotation test; therefore, unobstructed pictures are not available. In response to the reviewer’s comment, we have included experiment information for each sample in the pictures.

  1. Please check whether the sentences and words in the text are correct, such as whether ' Coreflooding ' should be changed to ' Core flooding '.

Response: Thank you for bringing this to our attention. We have carefully reviewed the entire manuscript, incorporating your suggestions to address grammatical and language-related errors. A professional English language editor also proofread the manuscript to ensure language accuracy. Additionally, we have updated 'Coreflooding' to 'Core flooding' throughout the manuscript.

  1. In order to highlight the main strengths of your research, extensive literature research should be conducted to shed light on important issues in oil extraction, some work in this area includes: The non-plane initiation and propagation mechanism of multiple hydraulic fractures in tight reservoirs considering stress shadow effects. Engineering Fracture Mechanics; Exploring the influence of rock inherent heterogeneity and grain size on hydraulic fracturing using discrete element modeling. International Journal of Solids and Structures.

Response: This comment is not for our manuscript. It seems to be an opinion on hydraulic fracturing in tight reservoirs.

Round 2

Reviewer 2 Report

No